# Psychological Well-Being, Self-Esteem, Quality of Life and Gender Differences as Determinants of Post-Traumatic Growth in Long-Term Knee Rotationplasty Survivors: A Cohort Study

**DOI:** 10.3390/children10050867

**Published:** 2023-05-12

**Authors:** Maria Grazia Benedetti, Ilaria Tarricone, Manuela Monti, Laura Campanacci, Maria Giulia Regazzi, Tiziano De Matteis, Daniela Platano, Marco Manfrini

**Affiliations:** 1Physical Medicine and Rehabilitation Unit, IRCCS Istituto Ortopedico Rizzoli, 40136 Bologna, Italy; mariagrazia.benedetti@ior.it; 2Department of Biomedical and Neuromotor Sciences, University of Bologna, 40126 Bologna, Italy; 3Bologna Transcultural Psychosomatic Team (BoTPT), Department of Medical and Surgical Sciences, Alma Mater Studiorum Bologna University, 40126 Bologna, Italy; ilaria.tarricone@unibo.it (I.T.); mariagiulia.regazzi@studio.unibo.it (M.G.R.); tiziano.dematteis.89@gmail.com (T.D.M.); 4Centre for the Promotion of Health and Psychological Well-Being, IRCCS Istituto Ortopedico Rizzoli, 40136 Bologna, Italy; monti.manuela@ior.it; 53rd Orthopaedic and Traumatologic Clinic Prevalently Oncologic, IRCCS Istituto Ortopedico Rizzoli, 40136 Bologna, Italy; laura.campanacci@ior.it (L.C.); marco.manfrini@ior.it (M.M.); 6Physical Medicine and Rehabilitation Unit, AUSL Romagna, 48100 Ravenna, Italy

**Keywords:** bone tumors, Van Nes rotationplasty, psychological well-being, quality of life, satisfaction

## Abstract

Rotationplasty (RP) is a special surgical technique for bone tumors of the lower limb and is the chosen procedure for children under 6 with bone sarcoma in the distal femur. Leg reconstruction results in an unusual aspect of the limb potentially giving life-long emotional outcomes, especially considering the young age of most RP patients. Although the high level of the quality of life of these patients has been previously reported, aspects related to long-term psychological well-being, self-esteem and life satisfaction, particularly regarding the gender, procreation and parenting, have never been explored. The aim of this study was to assess the general degree of psychological well-being of RP patients, with specific reference to gender, procreation and parenting. Twenty long-term RP survivors of high-grade bone sarcoma participated in the study. They were administered the following validated questionnaires: HADS for psychological well-being (degree of anxiety and depression), Temperament and Character Inventory (TCI), RSES for self-esteem, SF-36 for quality of life, SWLS extended to life satisfaction, and ABIS for body image integration. Data on education, marriage, employment and parenthood were gathered. All the scores obtained were very close to normal references. The only gender difference was found for the TCI Cooperativeness scale, which was higher in women than in men. A satisfactory psychological well-being in terms of both self-esteem and integration of the prosthetic joint limb into one’s body image, with relatively limited amount of anxiety/depression, good quality of life, and good temperament and character traits, was found. No major gender differences were reported.

## 1. Introduction

Rotationplasty (RP) is a very special surgical technique which, although not a true amputation, is often regarded as such. In its most frequent variant as treatment of bone sarcomas of the distal femur, it makes it possible to remove all thigh tissues, including the knee joint, while keeping the leg and foot innervations intact; the leg and foot are transplanted proximally after being rotated 180° and joined to the proximal femoral stump [1]. It is currently the procedure of choice for children under 6 years of age with bone sarcoma in the distal femur; however, it is also indicated for all age groups in cases of extremely extensive neoplasms covering the entire thigh and requiring removal of the femur, knee and all muscle groups en bloc [2,3].

Limb deformity has always been a cause for concern given the possible functional and psychological consequences, which can affect relationships, sexuality and parenthood. Actually, it has been amply demonstrated that, upon growing up, these patients have considerable functional reserves [4,5,6] and can lead completely normal lives, and many play high-impact sports [7,8]. Moreover, the psychological effects of the abnormal cosmetic appearance of the limb have been debated, concluding that in these patients the quality of life (QoL), as measured by SF36 is like that of a control population [9].

However, the various aspects of mental health and psychological well-being (measured in terms of anxiety and depression), satisfaction with the quality of life (also in relation to such fundamental aspects of adulthood as procreation and parenthood), and especially in relation to gender, have never been subject to a thorough, articulated investigation.

The aim of this study was then to use validated questionnaires to assess the general degree of psychological well-being of a sample of patients who underwent rotationplasty, with particular reference to gender, procreation and parenting. We hypothesized that in RP patients the degree of psychological adaptation in terms of self-esteem and integration of the prosthetic limb in the patient’s body image are specific and significant determinants of the above aspects.

## 2. Materials and Methods

### 2.1. Patients

The study population included all long-term survivors of rotationplasty performed at Rizzoli Orthopaedic Institute from 1986 to 2014. Twenty-eight patients who had undergone knee rotationplasty for high-grade bone sarcoma were invited to participate in the study. Ten patients living far from our institute were unable to come on the two days established for the assessment due to family or work-related issues. Twenty patients agreed to participate. The group included 11 women and 9 men, mean age 32.4 years (range 22–45), mean age at time of surgery 12 years (range 6–33), mean follow up after surgery 20.9 years (range 3–30). Two female patients underwent surgery, respectively, at the age of 33 and 31 years. No exclusions were made on the basis of age or nationality (as long as they could speak Italian). 

### 2.2. Assessment Tools

The instruments used are validated questionnaires, that, even at the initial administration, make it possible to assess the RP patient’s psychological well-being (in particular degree of anxiety and depression applying the HADS), quality of life (SF-36), self-esteem (RSES), response to emotional stimuli (TCI), life satisfaction (SWLS), and body image integration (ABIS). 

The HADS (Hospital Anxiety and Depression Scale) is a questionnaire specifically developed by Zigmond and Snaith [10] to identify states of anxiety and depression in patients suffering from organic diseases, with the exclusion of areas where psychological distress indicators (such as headache, insomnia and weight loss) could be symptomatic of the organic disease itself or a consequence of therapy. The instrument consists of 2 7-item scales, 1 assessing anxiety, the other depression, with scores ranging from 0 to 3 for each item. For each of the statements, the patient is asked which of four possible options best describes his or her emotional state. The patient answers each item using a 4-point (0–3) response scale; thus, the possible scores ranged from 0 to 21 for anxiety and 0 to 21 for depression. In the same clinical setting, analysis of the 2 subscale scores for a further sample showed that a score of 0 to 7 for either subscale could be deemed within the normal range, a score of 11 or higher indicated probable presence of a mood disorder, and a score ranging from 8 to 10 merely suggested the presence of the respective state.

The Temperament and Character Inventory (TCI) [11] is a battery of tests designed to assess differences between individuals with respect to specific parameters, called dimensions, so they can be used as a diagnostic tool in the clinical evaluation of psychiatric patients and thus assist in selection of the therapeutic strategy. The TCI takes into account seven major dimensions, divided into two domains, which are the main components of personality: (1) Temperament refers to the set of innate responses to emotional stimuli. It is partially inheritable, manifesting from birth and moderately stable throughout life. This is evaluated through 4 dimensions: “Novelty Seeking” (NS), “Harm Avoidance” (HA), “Reward Dependence” (RD) and “Persistence” (P) (rated on a 35-point scale for HA; 40-point scale for NS; 24-point scale for RD; 8-point scale for P); (2) Character concerns the individual ideas, goals and values that influence voluntary choices, intentions and the meaning of one’s life experience. It is moderately affected by sociocultural teachings and individual maturation. The three dimensions by which it is measured are: “Self-Directedness” (SD), “Cooperativeness” (C) and “Self-Transcendence” (ST). Each of these personality aspects interacts with the others, affecting one’s ability to manage their emotions and their susceptibility to behavioral disorders (rated on a 44-point scale for SD; 42-point scale for C; 33-point scale for ST).

The Rosenberg Self-Esteem Scale (RSES) [12] assesses how satisfied a person is with him/herself, whether he/she feels worthwhile, his/her level of self-respect and pride, or whether he/she feels worthless and no good at all. This is a self-administered test consisting of 10 items scored from 1 to 4 on a Likert scale. The maximum score is 40: values between 10 and 16 indicate low self-esteem, those between 17 and 33 indicate medium self-esteem, and those between 34 and 40 indicate high self-esteem. 

The Short Form-36 Health Survey (SF-36) questionnaire translated and validated in Italy for the adult population [13] was used both for its Physical Component Summary (PCS) scale and the Mental Component Summary (MCS), based on the following items: Physical Functioning (PF), Role-Physical (RP), Bodily Pain (BP), General Health (GH), Vitality (VT), Social Functioning (SF), Role-Emotional (RE), Mental Health (MH). The score of the scale has a range from 0 to 100; higher scores indicate a better function in the area.

To assess the overall satisfaction, the Satisfaction with Life Scale (SWLS) [14], a 5-item scale designed to measure global cognitive judgments of one’s life satisfaction, was used. Items are scored from 1 to 7 points where 7 corresponds to Strongly agree, 6—Agree, 5—Slightly agree, 4—Neither agree nor disagree, 3—Slightly disagree, 2—Disagree, 1—Strongly disagree. The total score indicates satisfaction as follows: 31–35 Extremely satisfied, 26–30 Satisfied, 21–25 Slightly satisfied, 20 Neutral, 15–19 Slightly dissatisfied, 10–14 Dissatisfied, 5–9 Extremely dissatisfied. In order to gain insights into the various aspects of life, the same scale was used to assess satisfaction in work, social life, sexuality, hobby, sport and psychological well-being.

The Amputee Body Image Scale (ABIS) [15] is a 20-item measurement tool developed to evaluate body image impairment among the population of amputees. The subjects answered questions about their perceptions and feelings on a 5-point Likert scale. Three of these questions are reversed-scored. The score ranges from 20 to 100. High scores indicate a body image disorder.

Finally, data on marital status, number of children, level of education and occupational status were also collected.

### 2.3. Statistical Analysis

Analysis of descriptive statistics was performed for each variable collected. Scores obtained from the questionnaires were analyzed by the Student’s *t*-test, comparing men vs. women, single vs. married, and people with only compulsory education vs. people with secondary education. Bivariate correlations were used to verify possible correlations between the scores obtained from the administered questionnaires. In the case of the TCI subscales, the Wilcoxon test was performed, and the confidence intervals were compared with those obtained by Fossati et al. [11].

The statistical programs used were SPSS version 24 (IBM Corp., Armonk, NY, USA). and JASP version 0.9.2.

### 2.4. Etical Approval

This is an observational study authorized by the Institutional Ethical Committee of Area Vasta Emilia Centro (N. 42/2017/OSS/IOR) and registered in the Clinicaltrial.gov (ID: 0010899). All patients signed an informed consensus before participating in the study.

## 3. Results

### 3.1. Demographic Patient Information

A total of 11 patients were single at the date of assessment (55%), 8 were married (40%) and 1 patient cohabitated with a partner. Eight patients had children (considering both males and females). A total of 3 patients had a primary school education (15%), 9 had a high school diploma (45%) and 8 had a university or higher degree (40%). Details of their occupational status are given in Table 1.

### 3.2. Assessment

Twenty patients were included in the study. One of the participants was not able to complete all the questionnaires. In addition, the scores of one of the SF-36 subscales (role and emotional states) were not reported by two patients, making it impossible to calculate their Mental Component Summary (MCS); one participant did not complete the ABIS and RSES scores. 

Table 2 shows the average TCI, SF-36, HADS, ABIS, RSES and SWSL scores.

The average HADS Depression and Anxiety scores were below the cut-off limit for a positive score for anxiety (score = 6) and depression (score = 5.6) where, for both scales, 7 is the cut-off for potential pathology, and 10 for a diagnosis.

The PCS (55.6 ± 5.9) and MCS (50.6 ± 6.9) scores were comparable with the average scores for the Italian population in the same age range (52.7 ± 7.7 for PCS and 47.6 ± 10.1 for MCS) [13].

The TCI does not have any true standard value; therefore, we deemed it more appropriate to compare the results obtained with a cohort of healthy subjects [11]. It should be noted that, in Fossati et al. [11], the values for all subscales were reported, except for Persistence, and thus we have no reference for this parameter (Table 2). More specifically, most of the values reported in the present study agree with those obtained in Fossati et al. [11], except for the items Reward Dependence and Cooperativeness, which in our sample appear to have lower scores.

With regard to ABIS, the average score achieved is 42.2 (±9.2), which falls within the middle of the 0–100 possible range.

The RSES score of 33.8 (±4.4) places it at the upper end of the medium self-esteem range.

Finally, the SWLS mean value was 26.4 (±4.8), which indicates that patients were satisfied with their life. With respect to the adjunctive items, work, social and friendship life, sexuality, hobby, sport and psychological well-being data obtained are reported in Table 3; patients report an average value of about 5 (i.e., modest agreement in all the items), with most of the patients scoring 6 or 7 in all the items. 

Student’s *t*-test evidenced that the MCS scores for SF-36 (i.e., the general measure of mental well-being, calculated using the scores of the subscales for vitality, social activities, role and emotional state, mental health) were significantly lower (indicating lower mental well-being) in unmarried patients as compared to those whose status was married or living with a partner (*p* = 0.041).

Scores on the SF-36 subscale for Physical Activity were lower (thus indicating lower physical well-being) in patients who had only completed compulsory education, whereas they were higher in those patients who had continued their studies (*p* = 0.05).

With regard to gender, only the scores for the TCI Cooperativeness scale were higher in women than in men (*p* = 0.04).

No other differences were found using the *t*-test. Bivariate correlations, used to verify the existence of relationships between the scores obtained, revealed that lower MCS scores (indicating lower mental well-being) were correlated with higher HADS-anxiety scores (Pearson Correlation coeff −0.536, *p* = 0.033), and lower self-esteem scores were correlated with higher HADS-depression (Pearson Correlation coeff −0.555, *p* = 0.017) and HADS-anxiety (Pearson Correlation coeff −0.659, *p* = 0.003) scores. 

Any other association was not significant.

## 4. Discussion

Rotationplasty is a rare surgical procedure but, from a functional point of view, it is exceptionally effective. Moreover, given its particular aesthetic aspect, numerous studies have focused on the satisfaction and quality of life of these patients [16,17,18,19,20,21,22]. Table 4 provides a review of the studies showing SF36 quality of life scores; in particular, Forni et al. showed that the values reported for both physical (PCS) and mental components (MSC) are in line with the reference normative population or even exceed it [16]. 

Our cohort falls within the range of the values reported in the literature with minimal differences for the MSC, which are slightly lower than those in other studies, and the PCS scores, which are slightly higher but still within the average values. In particular, if we look at the individual items, we find that the value for Mental Health is below the minimum (68.8 points) and Physical Role is well above (97.8) the maximum values reported by the other studies.

It is likely that a longer follow-up and a higher age of the patients at the time of the evaluation may have had some effect on these minor differences.

In a review of quality of life in patients affected by sarcoma [23], an overall improvement was found in the physical aspects of QOL over time but not in psychosocial function or mental health. In particular, psychosocial wellbeing was found to be poorer than that of the general population, with no differences among patients with amputation, limb-sparing surgery or rotationplasty.

To our knowledge, no studies have ever targeted the psychological profile in terms of self-esteem, anxiety/depression or personality profile, such as temperament and character, or has dealt with issues such as parenting and procreation, especially in relation to gender in RP patients. These aspects are even more important if we consider that most of the patients underwent surgery in childhood. In fact, the literature reports that in young patients who are long-term survivors of bone cancer, this may have a significant and long-standing impact on psychological adjustment that may include self-concept, depressive/anxiety symptoms and sexual functioning [23,24,25].

Previous studies on survivors of pediatric lower limb tumors reported that they could experience physical consequences that adversely affected self-image and could be affected by their emotional and skeletal maturity at the time of diagnosis and treatment [26]. Education is a predictor of employment and being married for the first time (and did not divorce). Other studies report that these patients show deficits in relation to employment and marriage [27,28], but no inter-patient differences were found in degree of education, occupational status, and self-image between amputees and those whose limb was spared [26]. Regarding sexual function, Barrera et al. [25] found that male survivors of lower limb bone tumors experience better sexual function than women survivors; however, survivors of limb-sparing surgery proved to be more affected by sexual problems, depressive symptoms and poor self-perception than patients who had undergone rotationplasty and amputees. Teall et al. [24] describes better sexual functioning and less depressive symptoms in bone tumor survivors, particularly males, while female survivors are identified as being at higher risk for adverse psychosocial outcomes and may benefit from early psychosocial intervention.

Veenstra et al. [22] reported that only one-third to one-half of a sample of 34 patients reported negative effects of the surgery on initiating social and/or intimate contacts, body image and sexuality. Rodl et al. [29] evaluated 22 patients, concluding that especially for the items partnership and sexuality, no differences were found between those with rotationplasty and the general population.

In our study, 85% of the patients had a high school or university education (compared to 57% in the Veenstra study [22]), 45% were married or cohabiting (compared to 33% reported by Veenstra), and only 10% were unemployed with no gender differences in this regard (1 man and 1 woman). A total of 40% of the patients had children, and 25% of these were women. Furthermore 47.5% (7/19 patients) reported to strongly agree and 26.3% (5/19 patients) to agree with the question: are you satisfied with your sexual life? These data show that, in a high percentage of cases, the patient undergoing rotationplasty can live a normal sexual life without gender differences; indeed, a number of patients did procreate. 

With regard to the temperament and character traits assessed in the TCI, most of the item scores match those of the healthy cohort reported in another Italian study by Fossati et al. [11], thus indicating that there are no relevant differences between the two populations. Notably, in the Reward Dependence and the Cooperativeness score, we found some overlap between the two populations, with lower values than in the reference cohort. Furthermore, a gender difference was reported, more specifically for the Cooperativeness value, which was higher in women than in men. These findings are difficult to interpret. It can be said that in previous studies, patterns of temperament and character were associated with post-traumatic stress disorder (PTSD) [30]; in particular, the item Cooperativeness was found to be lower in patients with PTSD and was considered to be a resilience factor for PTSD symptoms because it represents an individual resource to cope with trauma events. This is consistent with the observation that long-term survivors of malignant tumors in childhood can develop PTSD symptoms in adulthood, with a prevalence ranging from 6 to 22%. The experience of treatment for cancer in childhood is a complex event that may change children’s development, personality and relationships with their parents and peers [31].

In terms of psychosocial well-being, the anxiety–depression disorder measured by the HADS scale is, on average, below the pathological limit of 7 points even though, in some patients, values ranging between 8 and 10 points are suggestive of the presence of a depressive/anxious state. Since it is not possible to make comparisons with other populations of rotationplasty subjects, the only possible comparison is with populations of amputees for whom the values available from the literature vary rather widely: 7.07 ± 5.08 and 4.10 ± 3.77 [32] and 8.1 ± 2.3 and 8.9 ± 1.8 [33], respectively, for depression and anxiety.

Self-esteem, as measured by the Rosenberg Self-Esteem Scale, obtained a value of 33.8 points, borderline for high self-esteem, with a small number of subjects in the middle range for self-esteem disorders. This is particularly important considering that, given the bizarre conformation of the limb, the rotationplasty procedure was initially deemed psychologically difficult to accept in terms of self-image. However, more recently, authors [9] agree that giving the patients adequate information on the image–function relationship of the limb generally results in good patient acceptance of the surgery. This is confirmed by the good values obtained with the ABIS scale, which is designed to assess body image in amputees. On average, the population of patients who underwent rotationplasty reports a value of 42 points, which is significantly lower than the values reported by Gozaydinoglu et al. [34] in a population of transfemoral amputees (50.6 (16.6)) and those reported by Vouilloz et al. [35] in lower limb amputees at different levels (57.1 (16.9)).

Due to the specific features of the cohort population, the results obtained in the present research are generalizable only in relation to the cultural, social and clinical contexts of any other patients who have undergone the same intervention. Nevertheless, the study adds consistent knowledge about long-term QoL and psychological attributes of rotationplasty survivors.

We believe that having information about the psychological impact of rotationplasty over time, in particular with regard to gender, procreation and parenting, might be significant to families, patients themselves and those eligible for surgery.

Educational and psychological preparation for surgery may be beneficial in terms of postoperative recovery for most patients undergoing surgery [36], and in oncological patients, targeted intervention and programs of “prehabilitation” are recommended to improve general psychological health and behavioral recovery [37]. In the case of rotationplasty for oncologic indications where eligible patients are mainly children, counseling and psychological support impact the decision-making process and the rehabilitation project [38].

## 5. Conclusions

By means of specific and validated questionnaires, we addressed the quality of life and the general degree of satisfaction with psychological and mental well-being in a consistent sample of patients who underwent rotationplasty in childhood and were long-term survivors of bone cancer. We found a general satisfactory degree of psychological well-being in terms of both self-esteem and integration of the prosthetic joint limb into one’s body image, with a relatively limited degree of anxiety/depression, good quality of life both in psychological and motor terms, and lower temperament and character traits with respect to the reference population. No major gender differences were reported for the measures taken. In particular, the level of education, number of patients who formed a family and, especially, the number of women who procreated all suggest that this surgical procedure can result in very good psychological adjustment and a sense of existential fulfilment. In those patients showing poorer outcomes, there is a need to understand which determinants play the greatest role in the lower acceptance of the surgery and in the development of psychosocial well-being.

## Figures and Tables

**Table 1 children-10-00867-t001:** Sample characteristics.

Cases	*n* (%)
**Total**	20 (100)
Male	9 (45)
Mean age (±SD)	32.4 ± 7
**Marital status**	
Single	11 (55)
Married	8 (40)
Living together	1 (5)
**Had children**	8 (40)
**Education**	
Compulsory	3 (15)
High school	9 (45)
University degree and above	8 (40)
**Occupational Status**	
Artisan/retailer	2 (10)
Homemaker	3 (15)
Unemployed	2 (10)
Employee	6 (30)
Entrepreneur	2 (10)
Freelance worker	1 (5)
Blue-collar	1 (5)
Student	3 (15)
**Mean age of procedure** **(±SD)**	12 ± 7.5
**Mean time after the procedure (±SD)**	20.9 ± 8

**Table 2 children-10-00867-t002:** Mean scale values.

Scales	Mean Values (±SD)
**TCI** ^1^	
Novelty Seeking	24.1 (±3.9)
Harm Avoidance	19.6 (±2.9)
Reward Dependence	10.6 (±2.8)
Persistence	4.9 (±0.9)
Self-Directedness	27.4 (±3.5)
Cooperativeness	22.7 (±4.7)
Self-Transcendence	20.5 (±3.2)
**SF-36**	
PCS ^1^	55.6 (±5.9)
MCS ^2^	50.6 (±6.9)
**HADS** ^1^	
Anxiety	6.0 (±4.5)
Depression	5.6 (±3.6)
**ABIS** ^3^	42.2 (±9.2)
**Rosenberg Self-Esteem Scale** ^3^	33.8 (±4.4)
**SWLS**	26.4 (±4.8)

^1^ *n* = 19, ^2^
*n* = 17, ^3^
*n* = 18, Norms from Fossati et al. [11]: Novelty Seeking 28,6 (±5.6), Harm Avoidance18.36 (±7), Reward Dependence 16 (±3.35), Self-Directedness 25.39 (±7.6), Cooperativeness 31.43 (±5), Self-Transcendence 15.96 (±6.14).

**Table 3 children-10-00867-t003:** SWLS and adjunctive items.

SWLS Items	Mean	SD	Min	Max
In most ways my life is close to my ideal	5.1	1.3	2	6
The conditions of my life are excellent	5.3	1.3	2	7
I am satisfied with my life	5.9	0.9	3	7
So far I have gotten the important things I want in life	5.2	1.2	3	7
If I could live my life over, I would change almost nothing	4.9	1.8	1	7
** *Total SWLS* **	** *26.4* **	** *4.8* **		
**Adjunctive items**				
I am satisfied with my working life	5.4	1.6	1	7
I am satisfied with my social and friendship life	5.9	1.1	3	7
I am satisfied with my sex life	5.6	2.0	1	7
I am satisfied with the hobbies I practice	5.3	1.7	2	7
I am satisfied with the sport I practice	5.3	1.9	1	7
I am satisfied with my level of psychological well-being	5.7	1.5	2	7

**Table 4 children-10-00867-t004:** Literature review of SF36 data in rotationplasty patients. Data are expressed as mean (SD).

	n ^1^	FU	Age at Intervention	Age at the Assessment	SF36									
		years			PCS	MCS	Physical Functioning	Role Physical	Bodily Pain	General Health	Vitality	Social Functioning	Role-Emotional	Mental Health
Forni et al., 2012 [16]	20	17.2 (4)		26.9 (5.3)	53 (6.4)	52.4 (6)	89.3 (8.8)	81.3 (30.2)	88.2 (19.7)	83.6 (12.4)	71.8 (13.7)	81.2 (20.5)	83.3 (22.4)	80.8 (12.0)
Ginsberg et al., 2007 [17] *	4	9.6 (6.4)	14.5 (4.06)		52.7 (4)	52 (3.8)								
Hopyan et al., 2006 [18]	5	8.6 (3.5)	10.4 (4.8)		53.6 (3.7)	56.9 (2.5)								
Akahane et al., 2007 [19]	5	4.9	14.3 (8-14)				75 (13.2)	83.4 (14.4)	77.7 (20.1)	68 (19.7)	65 (18)	91.7 (14.4)	100 (0)	85.3 (12.9)
Gradl et al., 2015 [20]	12	14 (9)	19 (10)				80.4 (15.7)	78.1 (24.1)	74.1 (17.6)	71.8 (26.7)	75 (12.8)	98.9 (3.6)	88.2 (24)	89.6 (10.1)
Barrera et al., 2011 [21] **	6		11.6 (3.3)	25.1 (5)	48.3 (11.2)	55.6 (8.5)	71.1 (21.5)	78.6 (38.3)	74.9 (23.7)	74.2 (22.0)	64.46 (21.2)	86.6 (19.5)	73.8 (41.9)	
Veenstra et al., 2000 [22]	33	6.3 (1–11)					70.6	77.3 (30.9)	77.7 (20.5)	77.4 (22.7)	66.8 (21.9)	90.2 (17.6)	85.9 (27.7)	79.3 (17.4)
**Min**					**48.3**	**52**	**70.6**	**77.3**	**74.1**	**68**	**64.5**	**81.2**	**73.8**	**76.14**
**Max**					**53.6**	**56.9**	**89.3**	**83.4**	**88.2**	**83.6**	**75**	**98.9**	**100**	**89.6**
Present study	20	20.9 (8)	12 (5)	32.4 (7)	55.6 (5.9)	50.6 (6.9)	87.4 (12.8)	98.7 (5.7)	85.3 (14.8)	78.6 (14.7)	65.5 (13.6)	80.9 (20.1)	91.7 (19.3)	68.8 (14.5)

^1^ number of patients, * included in an amputated group, ** included in the whole sample of the study.

## Data Availability

Data supporting the findings of this study are available from the corresponding author upon reasonable request.

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
