# Peer review of "Psychological Well-Being, Self-Esteem, Quality of Life and Gender Differences as Determinants of Post-Traumatic Growth in Long-Term Knee Rotationplasty Survivors: A Cohort Study"

_children, 2023, doi:10.3390/children10050867_

Round 1

Reviewer 1 Report

Rotationplasty is a good alternative to amputation for specific indications. It is a surgical technique when it is well done gives good functional and psychological results with good socio-professional integration and a low complication rate. However, patients should be well prepared psychologically before and after surgery. Indeed, the psychological impact can be significant and the "chimerical" aspect of the limb can even lead to secondary amputations. For this reason, I think it will be worth adding a small paragraph in the discussion on the importance of psychological preparation and its effect on the psychological results of this semi-conservative surgical technique.

Reviewer 2 Report

Lines 145-146: delete statement: "The distribution of non-normal values was assessed." By itself, this statement is uninformative.

Lines 175-176: Legend is unclear.  Does "missing" mean "Had missing data"? Consider using the single superscript "1" instead of 1, 2, 3 here. Should this say "; Norms from Fossati [11]..."?

Multiple instances: replace "T-test for independent samples" with "Student's t-test" or just "t-tests". t-tests are implicitly for independent samples unless explicitly called "paired t-tests". Use hyphen consistently.

Lines 213-215: Report correlation coefficients r in addition to the p-values so the reader has a sense of the strength of these associations.

Table 4: replace "N.Pt" with "No. of Patients"; hyphenate "Functioning" properly; In legend state whether values are means and standard deviations or something else.

Line 319-321: Not sure what is the intent of this self-deprecating and poorly constructed sentence.  It unnecessarily "sells the study short."    I believe this is a well done study that adds significantly to what is known about long-term QoL and psychological attributes of such long term survivors. It is not reasonable to expect a large sample for this rare procedure.

Lines 323—328:  This is a long run-on sentence that is difficult to read and needs to be broken up  into several sentences.

Overall, the English usage is excellent.  Some minor suggestions:

Line 198: replace "lightly agree" with "modest agreement"

Line 201: change: "T-test for independent samples" to "Student's t-tests..."

Line 204: change "is" to "was".

Line 258: clarify what is meant by "being in the first marriage".

Line 285-286:  change "a little overlapping" with "some overlap"

Line 294: insert "can" after "childhood"

Line 297: change "pairs" to "peers"

Line 323: insert "here" after "used"

Change "court" to count". 
